# Benchmarking Eliminative Radiomic Feature Selection for Head and Neck Lymph Node Classification

**DOI:** 10.3390/cancers14030477

**Published:** 2022-01-18

**Authors:** Zoltan R. Bardosi, Daniel Dejaco, Matthias Santer, Marcel Kloppenburg, Stephanie Mangesius, Gerlig Widmann, Ute Ganswindt, Gerhard Rumpold, Herbert Riechelmann, Wolfgang Freysinger

**Affiliations:** 1Department of Otorhinolaryngology—Head and Neck Surgery, Medical University of Innsbruck, 6020 Innsbruck, Austria; zoltan.bardosi@i-med.ac.at (Z.R.B.); matthias.santer@tirol-kliniken.at (M.S.); marcel.kloppenburg@tirol-kliniken.at (M.K.); herbert.riechelmann@i-med.ac.at (H.R.); wolfgang.freysinger@i-med.ac.at (W.F.); 2Department of Radiology, Medical University of Innsbruck, 6020 Innsbruck, Austria; stephanie.mangesius@i-med.ac.at (S.M.); gerlig.widmann@i-med.ac.at (G.W.); 3Department of Radiation-Oncology, Medical University of Innsbruck, 6020 Innsbruck, Austria; ute.ganswindt@i-med.ac.at; 4Department of Psychiatry, Psychotherapy and Psychosomatics, Medical University of Innsbruck, 6020 Innsbruck, Austria; gerhard.rumpold@tirol-kliniken.at

**Keywords:** head and neck squamous carcinoma, lymph nodes, extracapsular spread, computed-tomography, radiomics, feature extraction, sparse discriminant analysis, recursive feature elimination, genetic algorithms

## Abstract

**Simple Summary:**

Pathologic cervical lymph nodes (LN) in head and neck squamous cell carcinoma (HNSCC) deteriorate prognosis. Current radiologic criteria for LN-classification are primarily shape-based. Radiomics is an emerging data-driven technique that aids in extraction, processing and analyzing features and is potentially capable of LN-classification. Currently available sets of features are too complex for clinical applicability. We identified the combination of sparse discriminant analysis and genetic algorithms as a potentially useful algorithm for eliminative feature selection. In this retrospective, cohort-study, from 252 LNs with over extracted 30,000 features, this algorithm retained a classification accuracy of up to 90% with only 10% of the original number of features. From a clinical perspective, the selected features appeared plausible and potentially capable of correctly classifying LNs. Both the identified algorithm and features need further exploration of their potential as prospective classifiers for LNs in HNSCC.

**Abstract:**

In head and neck squamous cell carcinoma (HNSCC) pathologic cervical lymph nodes (LN) remain important negative predictors. Current criteria for LN-classification in contrast-enhanced computed-tomography scans (contrast-CT) are shape-based; contrast-CT imagery allows extraction of additional quantitative data (“features”). The data-driven technique to extract, process, and analyze features from contrast-CTs is termed “radiomics”. Extracted features from contrast-CTs at various levels are typically redundant and correlated. Current sets of features for LN-classification are too complex for clinical application. Effective eliminative feature selection (EFS) is a crucial preprocessing step to reduce the complexity of sets identified. We aimed at exploring EFS-algorithms for their potential to identify sets of features, which were as small as feasible and yet retained as much accuracy as possible for LN-classification. In this retrospective cohort-study, which adhered to the STROBE guidelines, in total 252 LNs were classified as “non-pathologic” (*n* = 70), “pathologic” (*n* = 182) or “pathologic with extracapsular spread” (*n* = 52) by two experienced head-and-neck radiologists based on established criteria which served as a reference. The combination of sparse discriminant analysis and genetic optimization retained up to 90% of the classification accuracy with only 10% of the original numbers of features. From a clinical perspective, the selected features appeared plausible and potentially capable of correctly classifying LNs. Both the identified EFS-algorithm and the identified features need further exploration to assess their potential to prospectively classify LNs in HNSCC.

## 1. Introduction

In head and neck squamous cell carcinoma (HNSCC), both the presence and the localization of pathologic cervical lymph nodes (LNs) with or without extracapsular spread (ECS) remain important negative prognostic factors in terms of overall survival [1,2,3,4]. Faisal et al. observed in a review including 531 oral HNSCC-patients the clinical presence of pathologic cervical LNs to be a significant prognosticator affecting overall survival [3]. In terms of LN localization, Meccariello et al. explored the role of selective neck dissection on regional disease control in a retrospective study including 60 oropharyngeal HNSCC-patients and observed that involvement of LNs in the latero-cervical neck levels (i.e., level V) correlates with staging and prognosis; these LNs should therefore be considered specifically [4].

Current criteria to diagnose pathologic LNs or ECS in contrast-enhanced computer tomography scans (contrast-CTs) are shape-based [5,6,7]. Mainly, and without further quantification, head and neck radiologists primarily consider LN maximum diameter and margins [5,6,7]. However, significantly more quantitative information about shape, texture, and intensity is contained within contrast-CTs that could be exploited for aiding clinical decision-making.

Radiomic analysis [8,9] is an emerging, data-driven technique aiming at extracting, processing and analyzing quantitative image-based information to help answer particular clinical questions. The core idea is to treat medical imaging data of patients as data-mineable sources of information for diagnosis. The typical radiomic workflow starts with data acquisition as part of the clinical diagnostic process, followed by manual or semi-supervised segmentation of clinically interesting features within the radiologic data sets. These extracted features may aid clinical decisions by providing quantitative measures on base of the available imaging data. This analysis can provide appropriate clinical metrics for staging and prognosis by regression and classification of radiologic data.

The most challenging and time-consuming part of this approach is the choice of quantitative features that best comply with the diagnostic concepts of head and neck oncology. Typically, hundreds of candidate features are extractable from contrast-CT images—many of which are highly redundant and highly correlated (e.g., LN’s diameter and volume or uniformity and entropy) so that for clinical usefulness the most discriminative combination of features needs to be identified.

Feature extraction is a well-explored problem in machine learning and radiomics [10,11,12]. Howard et al. applied machine learning to identify HNSCC-patients with intermediate-risk who could benefit from primary concurrent radio-chemotherapy (RCT). In this cohort study including 33,526 HNSCC-patients RCT was recommended in 44–52% of the patients. Machine learning was found to bear potential to improve selection of intermediate-risk HNSCC-patients in need of trimodal therapy [12]. However, there is a fine distinction between aiming to apply machine learning to solve a classification task most accurately in a black-box manner and picking a subset of (raw) features that may work well in clinical prediction in combination with a more conventional human-driven statistical analysis of the problem at hand. The clear trend in the first approach is to skip hand-crafted low- and intermediate level features and to learn data driven representations with powerful models (e.g., Convolutional Neural Networks (CNNs) [13]. These algorithms can provide state-of-the-art accuracy with good generalization power, even with superhuman performance in some tasks [14,15], but are often sample-inefficient and their internal representations are hard to interpret [16]. Furthermore, the collection of a large number of supervised samples in medical datasets is often prohibitively expensive. All this impedes acceptance in the medical diagnostic domain. Radiomics, the second approach, uses simple but effective data-mining techniques that might not produce state-of-the-art accuracies. However, this second approach could provide insight into the structure of the problem of clinical image processing and clinically relevant feature selection [17].

In this retrospective data-driven cohort study problem-aware radiomic feature extraction and identification of “underlying” structures from typical small-scale clinical datasets were investigated. The specific objective of this study of this the clinical problem of LN-classification in HNSCC-patients, was to explore eliminative feature selection algorithms (EFS-algorithms) for their potential to identify as small as feasible sets of features and yet to retain as much clinical “accuracy” as possible.

## 2. Materials and Methods

### 2.1. Patient Population

This retrospective cohort study adhered to the “Strengthening the Reporting of Observation studies in Epidemiology” (STROBE) guidelines [18]. From 2008 until 2020 patients of the institutional head and neck cancer registry at the Department of Otorhinolaryngology, Head and Neck Surgery, Medical University of Innsbruck, Austria, that had incident, histologically confirmed, advanced HNSCC (UICC III or IV) were eligible. All eligible patients were treated with primary RCT. Contrast-CTs for staging (“staging-CT”), planning of RCT (“planning-CT”) and restaging after RCT (“restaging-CT”) were available for all eligible patients. Exclusion criteria were nasopharyngeal carcinoma, other cancers than HNSCC, or curative surgery as part of the first-line treatment.

From 1100 potentially eligible patients 823 did not meet the inclusion criteria. From the remaining 287 patients a representative random sample of 28 patients (~10%) was drawn with SPSS 27 (IBM, Armonk, NY, USA). The study flow diagram and patient inclusion, modified according to STARD criteria [19], is depicted in Figure 1.

The mean (±standard deviation) patient age was 67 ± 10 years, ranging from 42 to 87 years; seven patients were female. The mean time between diagnostic- and planning-CT acquisitions was 20 days (±11 days; range 6–87 days). The mean time-interval between planning- and restaging-CT acquisitions was 110 days (±26 days; range 41–87 days). Mean overall survival from initial diagnosis was 35 months (±32 months; range 2–87 months). The clinical characteristics of the 28 included patients are presented in Table 1.

### 2.2. Contrast-Enhanced Computed Tomography Scans

Staging- and restaging contrast-CTs adhered to the head and neck CT imaging protocols of the Department of Radiology (Medical University of Innsbruck) and were acquired with a Light Speed VCT or a Light Speed 16 CT scanner (GE Medical Systems, Vienna, Austria). The scan volume ranged from the frontal sinus to the upper mediastinum with a resolution of 512 × 512 pixels, 2 mm slice thickness, collimation of 24 × 1.2 mm, and 0.45 pitch. Sagittal and coronal images were reconstructed form the axial images. As contrast agent, Jopamiro 370 (Bracco Austria GmbH, Vienna, Austria) was administered intravenously adjusted to the patient’s body weight.

Planning CT scans were subsequently acquired at the Department of Radiation Oncology, Medical University of Innsbruck. For all included patients imaging protocols were strictly adhered to. CT scans were exported in DICOM-format from the IMPAX EE PACS system (Agfa HealthCare, Bonn, Germany) and the PowerChart Hospital Information System Cerner, Kansas City, MI, USA).

### 2.3. Segmentation and Classification of Lymph Nodes

All diagnostic-, planning- and restaging-CTs were imported to the segmentation software (Elements, BrainLab, Munich, Germany). For each patient the three largest LNs were manually segmented in all data sets slice-by-slice in the axial plane using the “paint on slices” tool provided by the software. All segmented LNs were examined by two experienced, board certified head and neck radiologists with more than 15 years of clinical experience in head and neck CT reporting. The segmented LNs were classified as “pathologic” (Figure 2), “pathologic with ECS” (Figure 3) or “non-pathologic” (Figure 4), complying with current CT reading criteria for LN-malignancy [5,6,7].

LNs were classified “pathologic” in staging- and planning CTs, if axial diameters were >10 mm, LN margins were poorly defined, capsular contrast agent enhancement and/or central necrosis was observed [5]. LNs were classified “pathologic” if in restaging-CTs (i.e., after primary concurrent RCT), the maximum short axis LN diameters was >10 mm, focal LN abnormalities (lucencies, contrast-enhancement or eccentric LN bulging) were observed, and/or if an increase of the maximum short axis LN diameter of >2 mm in restaging-CTs was observed [6] (Figure 2).

LNs were classified “pathologic with ECS” if in staging-, planning- or restaging-CTs, in addition to the criteria mentioned above [5,6], apparent fat/soft-tissue infiltration, infiltration of the sternocleidomastoid muscle, the internal jugular vein, or the carotid artery was observed [7]. Consequently, all LNs classified as “pathologic with ECS” were considered “pathologic”, too, but not all “pathologic” LNs were classified “pathologic with ECS” (Figure 3).

LNs were classified “non-pathologic” if in staging-, planning- or restaging-CT neither the criteria for malignancy [5,6] nor for ECS [7] were observed (Figure 4).

Since all patients enrolled in this retrospective cohort-study were treated with primary concurrent RCT no histological or cytological verifications of the segmented LNs were available, except for four patients diagnosed with carcinoma of unknown primary (Table 1). In these four patients, ultrasound-guided core-needle biopsies of only the largest suspect cervical LN were performed to histologically confirm the diagnosis of HNSCC. In the remaining patients the diagnosis HNSCC was histologically confirmed by obtaining tissue samples from the primary tumor during staging panendoscopy [20]. As recommended by the National Comprehensive Cancer Network [21] no additional ultrasound-guided core-needle biopsies of any of the suspect cervical LNs were performed in these patients. This diagnostic approach aimed at preventing the small yet existing chance of tumor seeding [22].

Consequently, the classification of the segmented LNs in “pathologic” (Figure 2), “pathologic with ECS” (Figure 3) and “non-pathologic” (Figure 4) by the two experienced head-and-neck-radiologists based on established criteria [5,6,7] was defined as reference.

### 2.4. Export of Segmented Lymph Nodes for Feature Extraction

All segmented LNs from all contrast-CTs with their respective classifications were exported from the BrainLab system in DICOM format. Segmentation boundaries were converted into binary volumes and saved as “Nearly Raw Raster Data” (NRRD) label volumes [23]; original CT images were saved as NRRD intensity volumes [23]. The open source PyRadiomics library [24,25] allows extraction of 120 different radiomic features, including intensity based, shape-based and texture-based features for each segmented LN (Appendix A).

### 2.5. Feature Selection

Various supervised and unsupervised wrapper-type feature selection variants [26] to reduce the number of features (without recombination into new features) were explored. This allows easier interpretation of the selected features by avoiding feature extractors that use dense recombination of features (e.g., principal component analysis or factor analysis [27]).

The primary aim of this retrospective cohort study was to identify at least one EFS-algorithm for each of the LN-classifications “pathologic” (Figure 2), “pathologic with ECS” (Figure 3) and “non-pathologic” (Figure 4), capable of maintaining a classification accuracy >80% with a reduced number of active features *p* to ≈10.

The standard radiomic features extracted from CT datasets tend to be highly correlated and are prone to multi-collinearity (e.g., [28]). This may hinder the effectiveness of choosing features with a generally good performance as diagnostic radiomic signals. From a machine-learning perspective, a high correlation between clusters of features can have a negative effect on some classifiers [29] and their associated supervised feature reduction techniques [30].

A set of simple linear discriminant analysis (LDA) classifiers [31] was trained on all available features for all available binary labels in the dataset to establish a baseline of feature selection. The motivation for this is twofold: for small datasets in which the number of data points is only a few times the number of features, almost any further increase in model complexity can lead to overfitting [17]. Additionally, linear classifiers have a straightforward metric to estimate relative feature importance and are simple to use within wrapper-type feature selection algorithms. Neither generalization performance nor absolute accuracy of the classifiers are considered at this stage. Simply, different feature selector candidates are evaluated on base of their relative performance.

The performance of a feature selector S_a_ is considered to be better than that of a feature selector S_b_, if for a fixed *p* << |*Ƒ*|, |*Ƒ*| being the cardinality of *Ƒ*, the set of all features, the classification accuracy of S_a_ on the feature set provided by *Ƒ*_Sa_ is higher than the classification accuracy of S_b_ on the feature set *Ƒ*_Sb_ provided by S_b_; all other input remains fixed. The question to answer is how to efficiently select a subset of features *Ƒ*_s_ from *Ƒ* such that *Ƒ*_s_ still retains as much of the original classification accuracy as possible.

### 2.6. Candidate Feature Selection Algorithms

Two main classes of algorithms were considered: the first class were explicit parametric algorithms, where the exact number of the desired dimension count is set. Examples for explicit parametric algorithms are recursive feature elimination (RFE), subclass discriminant analysis (SDA) and genetic algorithm (GA). The second class were implicit parametric algorithms, where a specific hyperparameter determines the final dimensionality based on the data. Examples for implicit parametric algorithms are highly correlated feature removal (HCFR) and recursive feature elimination with cross-validation (RFE-CV). Additionally, random feature selection [10,11,17] was evaluated.

All supervised techniques use an LDA classifier at each step to determine the achievable accuracy and the relative importance of each independent feature. Given the model weights {*w*_i_} and input features {*x*_j_}, both ∈ℝi, label y is given by:(1)y=f(ω·x)=(∑jωjTxj)
(2)f(x)={1 if x>S,S being a sufficient treshold 0 otherwiseThe (relative) importance of the feature f_j_ ∈ *Ƒ* is assumed to correspond to |ω_j_|.

#### 2.6.1. Highly-Correlated-Feature Removal (HCFR)

Highly-Correlated Feature Removal (HCFR) is an unsupervised, implicitly parametrized technique, that samples a set of minimally correlated features. A maximum correlation threshold S is given as a parameter defining the maximum allowed correlation between any two features in the selected feature set. The threshold is enforced by agglomerative clustering [32], with an affinity between features $f_{p},\;f_{q}$, *q* ∈ *Ƒ* defined as aff(f_p_, f_q_): = 1 − abs(corr(f_p_, f_q_)). Then the selected feature set is assembled by sampling a random feature from each cluster. The resulting feature set will be stochastic and decorrelated within the constraints of the original data and feature-set.

#### 2.6.2. Recursive Feature Elimination (RFE)

RFE is a supervised, explicit, greedy feature selection technique that iteratively decreases the number of features by repeatedly fitting the classifier and removing the least important *n* features, typically *n* = 1, until the target number of features, *p*, is reached [10,11]. When used with linear classifiers such as support vector machines (SVM) or LDA as classifiers in form (1), relative feature importance is determined by the magnitude of weight ω_j_ associated with feature *i* [10,11].

Combining RFE with *k*-fold cross-validation (FRE-CV) may be useful for finding features that are more generic. At each evaluation step a complete *k*-fold cross validation of the classifier is executed with the current set of features. This yields means and standard deviations of classification accuracies as well as feature weights from *k* executions per step, reducing the effect of singular weight inconsistencies and allowing stable evaluation of step accuracy. RFE is considered a supervised but implicitly parametrized feature extraction algorithm, because the optimal number of features is selected based on a predefined accuracy/feature count trade-off predefined by the algorithm [10,11].

#### 2.6.3. Sparse Discriminant Analysis (SDA)

Clemmensen et al. proposed sparse discriminant analysis (SDA) [33], which applies elastic net regularization on the weight matrix *β* in the form:(3)$$min_{\beta }\left\{\|\gamma -Χ\beta {\|}^{2}+\lambda \|\beta {\|}_{1}+\gamma \|\beta {\|}^{2}\right\}$$

Hyperparameters *λ* and *γ* control the accuracy-sparsity trade-off and enforce sparsity of the features for larger values; X is the data vector. Careful tuning of these hyperparameters ensures that exactly *p* features have non-zero weights. LDA classifiers are trained on the features selected in the SDA analysis to provide the final classification accuracy. In the proposed form SDA*p* can be seen as an explicit and supervised feature reduction method.

#### 2.6.4. Genetic Algorithms (GA)

GA are often applied to solve the problem of feature selection (e.g., [34]); and are a combinatorial approach for optimization problems where from the original set of features *Ƒ* a subset *P* ∈ *Ƒ* is sought and two main objectives are expected to be optimal simultaneously: the first objective depends on the number of features, c_1_(|*P*|), while the second measures the performance of the classifier when using the selected subset of features c_2_(*P*, *x*, *y*). This is typical for multi-objective genetic algorithms (MOGA) [35,36].

When using a weighted-sum combination c = ∑_i={1,2}_ λ_i_c_i_(·) for some user-specified importance factors λ_i_, i.e., setting a limit on the number of features *p*, the problem simplifies and can be solved with single-objective GA optimization. Following a weighted-sum representation with λ_1_ = λ_2_ = 1, c_1_(|*P*|): = min(|*P*| − *p*, 0), and c_2_(*P*, *x*, *y*): = the negative balanced accuracy (BACC) reached by LDA classification on the dataset x with labels y when using features *P*; c(·) is minimized. This ensures Pareto domination [37] to feature configurations with smaller feature counts given no advantage in terms of BACC as long as the feature count is over the targeted limit. GA*p* can be considered a “soft” explicit, supervised feature reduction algorithm, as the desired number of features *p* is explicitly defined, but the final number of proposed features may deviate from the exact value of *p*.

#### 2.6.5. Random Feature Selection (RND)

Random feature selection is a trivial yet sometimes surprisingly effective explicit unsupervised algorithm to reduce the number of input features for classification [10,11,17]. A uniformly randomly selected subset of size *p* from the set of all input features *Ƒ* is used for classification. Although its variance is large, this method can give a good reference to evaluate more advanced feature selectors and can provide an efficient best-of-N type feature selection algorithm when enough samples are considered [10,11].

#### 2.6.6. Dual-Phase Feature Elimination

Additionally, dual-phase feature elimination was also considered: the first phase reduces the feature dimensions to a moderate number (~50) using non-combinatorial techniques such as SDA or RFE. The second phase uses genetic optimization for further reducing feature dimensionality to the desired target count. The two versions are termed SDA*q*GA*p* and RFE*q*GA*p*.

#### 2.6.7. RFE with Highly Anti-Weighted Feature Elimination Filter (HAFF) Preprocessing

While in general the heuristic of linear weight as feature importance can be considered effective, it has some drawbacks [38]. When the original feature set has subsets of highly correlated feature pairs (*f*_i_, *f*_j_), *ω*_ij_ = corr(*f*_i_, *f*_j_) > T, T ≈ 0.95, and heavily anti-weighted feature pairs (*f*_k_, *f*_l_), *ω*_kl:_ = corr(*f*_k_, *f*_l_) < −T, T ≈ 0.95 (*ω*_kl_ << 0 << *ω*_ij_) linear classifiers may reach degenerate solutions. Highly anti-weighted feature pairs distort the feature importance statistics, because at least one of the feature pairs (i, j) is typically much less important for a good classification than its feature magnitude would imply and a more important feature might be removed from the set. RFE is particularly vulnerable to this phenomenon; removing highly anti-weighted feature pairs in a preprocessing step can have positive effects on candidate feature selection methods. We propose RFE-HAFF as an alternative form of RFE. At each feature removal step, instead of the least important feature, the highest anti-weighted features (HAF) with an absolute correlation above a specified threshold S are removed. RFE-HAFF terminates when all HAFs are eliminated and feature elimination proceeds in the usual manner. Therefore, HAFF performs a greedy, unsupervised elimination of some features at the preprocessing stage prior to executing the main feature selector algorithm.

### 2.7. Candidate Feature Selection Procedure

First, LDA performance was established as a baseline against which to assess the performance of the different feature reduction methods. LDA was trained on the LN-classifications “pathologic”, “pathologic with ECS” and “non-pathologic” for all 252 LNs with all 120 features.

In the first evaluation step, the performance of the classifiers was evaluated after an approximate 50 percent reduction in feature count. Both definite- and indefinite feature selectors were executed with *p* = 51 as the desired feature count. Final results may differ for indefinite selectors. The exception is RFE-CV, where *p* does not need to be specified. For two-phase algorithms a 50 percent reduction in the feature count was targeted in the first phase, followed by an approximate 80 percent reduction (with a maximum of *p* = 10) in the second phase.

## 3. Results

### 3.1. Classification of Lymph Nodes and Feature Extraction

A total of 28 patients and with 252 LNs were classified based on established criteria [5,6] by two independent head and neck radiologists (Table 1). Of these, 182 LNs were classified as “pathologic” [5,6] and further 52 of these were classified as “pathologic with ECS” [7]. No new LNs with ECS developed neither from “pathologic” LNs between acquisitions of staging- and planning-CTs nor between planning- and restaging-CTs. Seventy LNs did not meet the criteria for malignancy [5,6] or ECS [7] and were labeled “non-pathologic”.

From the 252 segmented LNs a total of 30,240 features were extracted. 4788 were first order statistics features, 4032 three-dimensional shape-based features, 2520 two-dimensional shape-based features, 6048 grey level co-occurrence matrix features, 4032 grey level run length matrix features, 4032 grey level size zone matrix features, 1260 neighboring gray tone difference matrix features, and 3528 gray level dependence matrix features (Appendix A).

### 3.2. Candidate Feature Selection Algorithms

For each of the LN-classifications “pathologic”, “pathologic with ECS” and “non-pathologic” results are presented in the following figures. In each figure, the boxplots show the distribution of the BACC of LDA classifiers trained on the features selected by the used selection algorithm. Each figure shows the performance of the baseline LDA classifier trained on all available features as a dash-dotted horizontal line for reference. The naming convention for single phase algorithms and for dual-phase algorithms is FirstphasealgorithmFirstphasetargetfeaturecount_(haff|nohaff)_ (|BACC) and FirstphasealgorithmFirstphasetargetfeaturecountSecondphasealgorithmSecondphasetargetfeaturecount_(haff|nohaff)_BACC, respectively.

First phase algorithms refer to RFE, SDA and RNF where the target feature count defines the number of target features to be reached by the first phase algorithm. Labels “haff” or “no-haff” indicate whether highly anti-correlated feature filtering heuristic was applied prior to the first phase or not.

### 3.3. EFS-Algorithm for LN-Label “Pathologic”, “Pathologic with ECS”, and “Non-Pathologic”

Balanced accuracy distributions of EFS-algorithms trained and evaluated on the LN-label “pathologic” (Figure 5), “pathologic with ECS” (Figure 6) and “non-pathologic” (Figure 7) are depicted below. For each of the figures, the BACC of the LDA classifier without featre count reduction is shown as dashed-dotted green horizontal line, for reference. For additional detail please refer to the individual figure legends.

From a clinical perspective, the selected features for the LN-label “pathologic with ECS” appeared plausible and potentially capable of classifying LNs (Table 2).

## 4. Discussion

In HNSCC both the presence and the localization of pathologic LNs with or without ECS remain an important negative prognostic factor [1,2,3,4]. The latter was observed to correlate with the specific LN involvement of the laterocervical neck level (i.e., level V) [4]. Current criteria to diagnose pathologic cervical LNs in contrast-CTs are primarily shape-based [5,6,7]. Radiomic analysis provides numerous, additional quantitative information, including texture and intensity features [8,9]. One of the most challenging parts of radiomic analysis is the choice of features that best comply with the diagnostic concepts explored [10,11]. Machine learning has been applied for this task in a large patient cohort study recently [10,11,12] and it might have the potential to improve intermediate-risk HNSCC-patient selection for trimodal therapy [12]. In this retrospective cohort-study EFS-algorithms were explored for their potential to correctly classify LNs with sets of features as compact as possible and yet retaining as much clinical accuracy as possible.

The primary aim was to identify at least one EFS-algorithm for each of the LN-classifications (Figure 2, Figure 3 and Figure 4), which was capable of maintaining a classification accuracy >80% with a reduced number of active features *p* ≈ 10. The observations of the present study suggest for LNs classified as “pathologic” the combination of RFE and GA, for “pathologic with ECS” and “non-pathologic” the combination of SDA and GA to be the potentially most useful EFS-algorithms, retaining a diagnostic accuracy of >80% (Figure 5), 90% (Figure 6) and 83% (Figure 7), respectively.

There are several different aspects to assess the performance of the different EFS-algorithms against each other. Probably the most important factor is the BACC for the classifier at a given number of features to be retained. The results show that for most cases decreasing the number of input features decreases the BACC of the LDA classifier. It is clear that reducing features to 10 percent of the original amount is a much harder task and only well-engineered feature selectors can significantly outperform random selection in this range. The advantage is not that obvious for the middle reduction range (50 percent feature reduction). While in theory due to the peaking phenomenon [10,11] the classification performance may not reach its maximum when all features from *Ƒ* are used, on our dataset LDA has achieved the best classification accuracies for all labels when all extracted radiomic features were available to it. Iterative feature removal algorithms have shown a monotonic decrease in classification accuracy.

It is evident from the results that the unsupervised, implicitly parametrized HCFR (0.303) algorithm had an underwhelming performance even when compared to random feature selection in the 10 percent range. Label awareness seems to be crucial to find the most discriminative features.

The cross-validation variant of RFE (RFE-CV) performed inferior to dual-phase GA variants by frequently yielding inferior accuracies with more features than the competing methods [10,11,38].

When inspected in the medium range of feature reduction (approx. 50 features, or 50 percent feature reduction for our study), all single-phase variants (SDA, RFE, GA) consistently seem to outperform random feature selection for all labels. However, the difference is relatively small and overall no clear advantage could be observed for any variant. In this evaluation regime, the HAFF heuristic was inconclusive; different trends were observed for different label types.

For the 10 percent features to be retained (i.e., 10 features in the present study) the greedy selection strategy of RFE, a single-phase algorithm, seems to break down performing similar to or worse than RND. Single phase algorithms SDA and GA on the other side produce robust results. The HAFF heuristic significantly improves RFE’s performance on two of the three available label sets (“pathologic with ECS” and “non-pathologic”), but still cannot reach the performance of SDA, GA, or dual phase methods.

GA10—as a single-phase algorithm—provides a robust performance in this range. Slight improvements can be observed when using the dual SDA50GA10 variant, which reduces the variance of the achieved classification accuracy and may reach higher absolute accuracies with repeated executions.

Overall—if enough computational power and time are available—the combinatorial optimization with genetic algorithms, GA*p*, seems to be the simplest and most robust approach for feature selection. This is not surprising due to the combinatorial nature of the problem. In practice, though, a dual phase algorithm that first reduces the number of initial features to an intermediate range (50 percent) and then starts a combinatorial search on this reduced set can be a practical and efficient approach. In these scenarios, SDA seems to outperform RFE for most of the cases.

From a clinical perspective the selected features especially for the LN-label “pathologic with ECS” appeared plausible and potentially capable of classifying LNs. The SDA50GA10_HAFF_BACC EFS-algorithm most frequently selected shape, intensity and texture as individual features from 100 repetitions. These specific features (Table 2) should further be explored for their potential to prospectively classify LNs as pathologic with ECS.

Certain limitations of the present study need to be addressed. Firstly, this was a retrospective study executed on a single, moderately sized clinical cohort, with only three different labels. From the 1100 potentially eligible patients only 287 met the inclusion criteria. In addition, to reduce the high workload of manually segmenting LNs the sample size was reduced by randomly sampling 28 patients (Figure 1 and Table 1). More data diversity would allow stronger conclusions about the general differences in the power of the evaluated feature selectors. The complexity of the classification model was kept as low as possible (linear) for the study in order to improve clinical applicability. Changing model complexity may affect both relative and absolute performance of the benchmarked methods.

Secondly, all patients enrolled in this study were treated with primary concurrent RCT (Table 1). Histopathologic confirmation of HNSCC was performed on tissue samples from the primary tumor staging panendoscopy [20] without additional ultrasound-guided core-needle biopsies. Four patients diagnosed with carcinoma of unknown primary (Table 1) received an ultrasound-guided core-needle biopsy of the largest suspect cervical LN to confirm diagnosis HNSCC.

Although this diagnostic approach is in line with the recommendations by the National Comprehensive Cancer Network [21], from a research perspective a histological or cytological verification of the segmented LNs would have improved the study’s quality. However, the risk of bleeding for ultrasound-guided core-needle biopsies is ~1% with a risk of tumor seeding of ~0.001% [22]. Moreover, three ultrasound-guided core-needle biopsies per patient and CT-scan would have been required to define the histopathologic result as reference method, multiplying the risk of adverse events. Lastly, classification “pathologic with ECS” (Figure 3) is frequently based on established radiologic criteria [7]. ECS can be confirmed histopathologically by removing the suspect LN in toto (e.g., via selective neck dissection); ultrasound guided core-needle biopsies cannot provide this information.

Ultrasound-guided fine-needle aspirations with known significant lower risk of bleeding and tumor seeding [22] of suspect cervical LNs was discussed for the patients of this study. While the number of ultrasound-guided fine-needle aspirations is the same, non-diagnostic aspirations of up to 7% and inconclusive cytological findings of up to 12% have been described and make this method less reliable a reference method [39].

Consequently, in this study, ultrasound-guided core needle biopsies or fine-needle aspirations were not used; instead, classification of segmented LNs as “pathologic with ECS” (Figure 2), “pathologic” (Figure 3) and “non-pathologic” (Figure 4) was completed by two experienced readers in head-and-neck-radiology based on established criteria [5,6,7] and was defined as the reference method. While there might be a risk of incorrect classification this was considered the safest approach for the patients.

It is to be emphasized that only HNSCC-patients treated with primary concurrent RCT were enrolled in this retrospective cohort study. Consequently, translating the findings of this study to HNSCC patients treated primarily with surgery needs great caution. Addressing this limitation in future studies appears necessary. Including patients undergoing surgical treatment was also considered for this study. However, during the conceptualization of the study it became clear that it might be difficult to identify resected LNs with preoperatively segmented LNs. Limiting the number of potentially segmented LNs might be a possible solution to this dilemma, e.g., by limiting patients to be included to HNSCC-patients with LN persistency after primary radiochemotherapy who frequently present one single persistent LN [40]. Unfortunately, this reduces the number of potentially eligible patients and therefore and does not allow conducting sufficiently powered studies. Consequently, in the present study the decision was made to include HNSCC-patients treated with primary concurrent RCT only.

In terms of external validity, the number of studies exploring the capability of radiomic analysis in classification of LNs in HNSCC-patients is sparse [41]. To best of our knowledge, only one similar study has been previously performed [41] that identified 89 features to differentiate pathologic from non-pathologic LNs and four features to differentiate pathologic LNs with ECS from pathologic LNs without ECS. This study employed magnetic resonance imaging of the neck, hampering comparability with our study [41]. In addition, no EFS-algorithms were explored. Even if a diagnostic accuracy of >80% was reported, a total of 89 features to correctly classify LNs appears too complex for daily clinical practice [41].

## 5. Conclusions

In conclusion, SDA, GA and their combinations are algorithms that can reduce the number of features significantly, while retaining sufficient discriminative power even for clinical datasets with small sample sizes and many highly correlated features. These methods might provide the necessary tools to prune abundant radiomic features prior to a more thorough manual analysis.

## Figures and Tables

**Figure 1 cancers-14-00477-f001:**
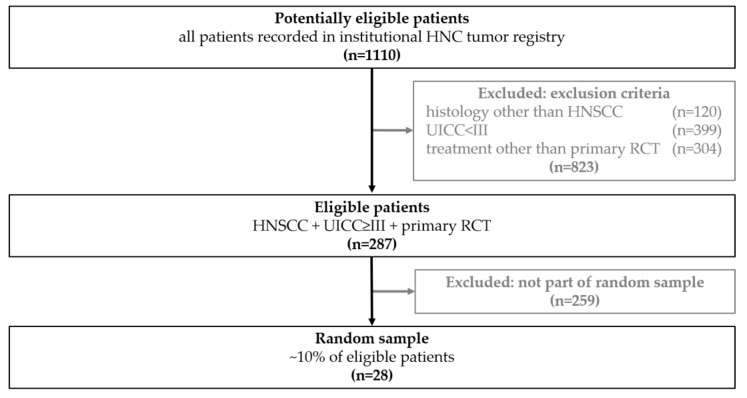
Study flow and patient inclusion modified according to STARD criteria [19]. A total of 1110 patients were potentially eligible, of which 823 did not meet the inclusion criteria. Of 287 eligible patients a representative random sample of 28 patients (~10%) was drawn. The clinical characteristics of the 28 included patients are presented in Table 1.

**Figure 2 cancers-14-00477-f002:**
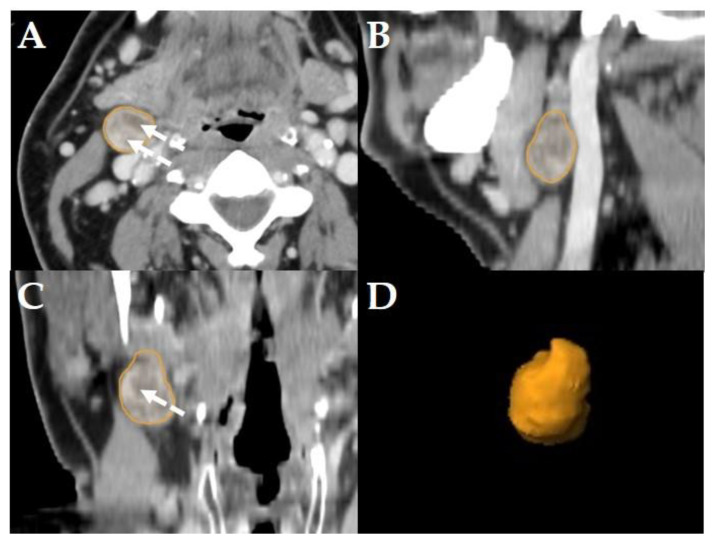
Example of a LN classified as “pathologic” in a staging-CT of a 55-year-old, male HNSCC patient with a tumor of the oral cavity staged cT4a cN2b cM0. Original segmentation in the axial plane (**A**); sagittal (**B**) and coronal (**C**) reformatted views and three-dimensional rendering (**D**) of the LN are provided by the software. Dashed arrows show central necrosis; no soft tissue infiltration and irregular LN capsule was observed.

**Figure 3 cancers-14-00477-f003:**
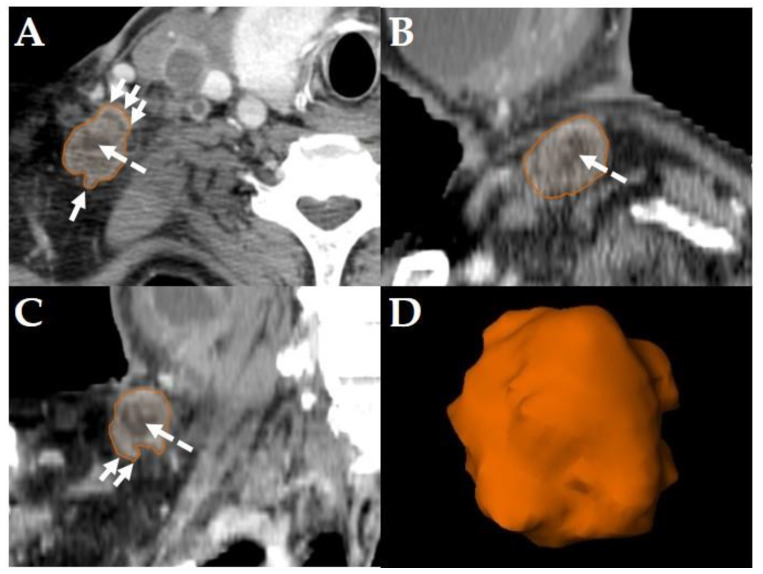
Example of a LN classified as “pathologic with ECS” in a Staging-CT of a 53-year-old, male HNSCC-patient of the oral cavity staged cT2 cN3b cM0. Manual segmentation in the axial plane (**A**); sagittal (**B**), coronal (**C**) reformatted views and three-dimensional rendering (**D**) of the LN are provided by the software. Solid arrows show soft tissue infiltration and an irregular LN capsule. Dashed arrows show central necrosis.

**Figure 4 cancers-14-00477-f004:**
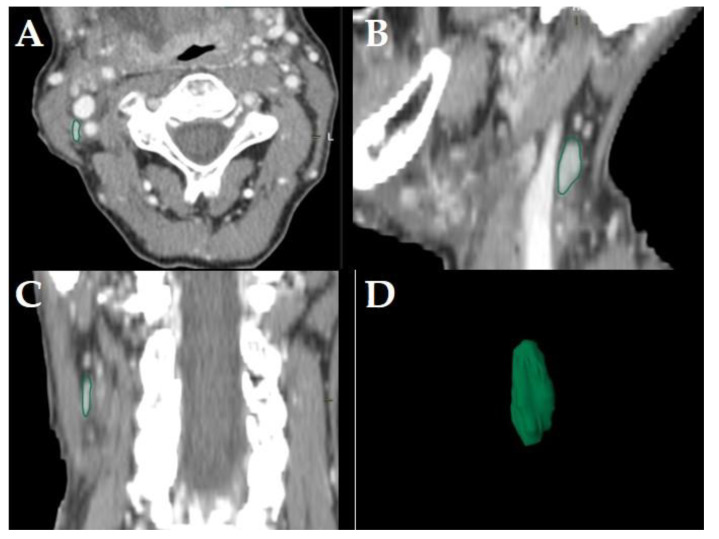
Example of a LN classified as “non-pathologic” in a staging-CT of a 65-year-old, female HNSCC-patient with a tumor of the oral cavity staged cT4a cN2c cM0. Manually segmented LN in the axial plane (**A**); sagittal (**B**), coronal (**C**) reformatted views and three-dimensional rendering (**D**) of the LN are provided by the software. Neither central necrosis nor soft tissue infiltration nor irregular LN capsule were observed for this LN.

**Figure 5 cancers-14-00477-f005:**
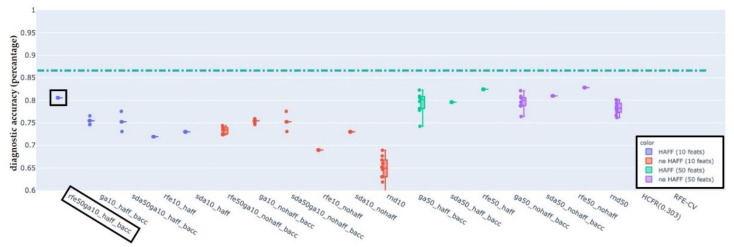
Balanced accuracy distributions of EFS-algorithms trained and evaluated on the LN-label “pathologic”. For reference, the BACC of the LDA classifier without feature count reduction is shown as a dash-dotted green horizontal line (at a value of 0.87). The combination of RFE and GA (black box) appeared to be the potentially most useful EFS-algorithms, retaining a diagnostic accuracy of >80% with only 10% of the features.

**Figure 6 cancers-14-00477-f006:**
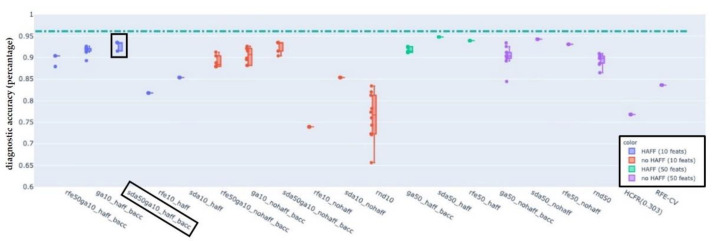
Balanced accuracy distributions of EFS-algorithms trained and evaluated on the LN-label “pathologic with ECS”. For reference, the BACC of the LDA classifier without feature count reduction is shown as a dash-dotted green horizontal line (at a value of 0.96). The combination of SDA and GA (black box) appeared to be the potentially most useful EFS-algorithms, retaining a diagnostic accuracy of >90% with only 10% of the features.

**Figure 7 cancers-14-00477-f007:**
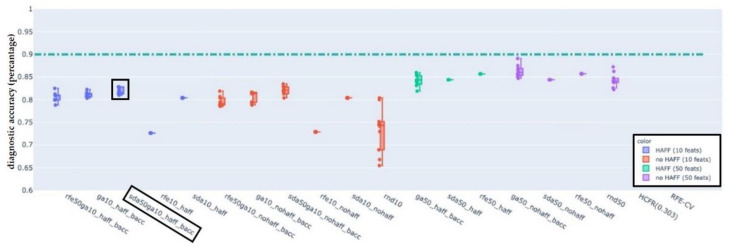
Balanced accuracy distributions of EFS-algorithms trained and evaluated on the LN-label “non-pathologic”. For reference, the BACC of the LDA classifier without feature count reduction (dash-dotted green horizontal line at 0.90). The combination of SDA and GA (black box) appeared to be the potentially most useful EFS-algorithms, retaining a diagnostic accuracy of >83% with only 10% of the features.

**Table 1 cancers-14-00477-t001:** Clinical characteristics of the 28 included HNSCC-patients.

Sex	male	20
	female	8
Age	≤50 years	1
	51–60 years	9
	61–70 years	9
	71–80 years	8
	≥80 years	2
p16 Status ^1^	Negative	14
	Positive	6
	Unknown	8
Primary Tumor Site	oral cavity	6
	oropharynx	6
	hypopharynx	6
	larynx	6
	CUP ^2^	4
Clinical T-stage	T1	0
	T2	7
	T3	6
	T4	11
Clinical N-stage ^3^	N0	0
	N1	4
	N2	19
	N3	5

^1^ p16 was also assessed in several non-oropharyngeal HNSCC; ^2^ carcinoma of unknown primary; ^3^ initial N-stage and label of LN not necessarily match, since the head and neck radiologists were blinded to the written report.

**Table 2 cancers-14-00477-t002:** The 10 most frequently selected features by the EFS-algorithm SDA50GA10_HAFF_BACC after 100 repetitions for classification of LNs labeled “pathologic with ECS”.

Type ^1^	Name ^2^	Number ^3^
Shape	surface-to-volume ratio	100/100
Intensity	first order statistics median	80/100
	first order statistics skewness	100/100
Texture	gray level co-occurrence matrix inverse difference moment	95/100
	gray level co-occurrence matrix inverse difference normalized	45/100
	gray level dependence matrix low gray level emphasis	75/100
	gray level run length matrix long run gray level emphasis	30/100
	gray level run length matrix run entropy	70/100
	gray level run length matrix short run emphasis	35/100
	gray level size zone matrix zone entropy	100/100

^1^ Type refers type of radiomic feature (i.e., shape, intensity or texture); ^2^ Name refers to the specific name of the individual feature; ^3^ Number refers to number of times features were selected per 100 repetitions.

## Data Availability

The data presented in this study are available on request from the corresponding author.

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
