# Peer review of "Benchmarking Eliminative Radiomic Feature Selection for Head and Neck Lymph Node Classification"

_cancers, 2022, doi:10.3390/cancers14030477_

Round 1

Reviewer 1 Report

The authors present a study on radiomics using 30000 features to evalaute lymph nodes in head and neck carcinoma patient cohort.

It is clinically very interesting and valuable topic. The results are nicely shown with numerous figures.

Design of the study misses what is used as a golden standard. Is there a histological or cytological verification of the results? Please expand this issue and explain it in details.

Discussion is quite short and should be expanded.

Author Response

Dear editors, dear referees,

thank you very much for the extension of the revision deadline over the Christmas holidays. All the authors very much appreciate the patience of both, editors and referees. The authors very much appreciate the constructive and positive comments of both referees about the manuscript “Benchmarking eliminative radiomic feature selection for head and neck lymph node classification” (ID cancers-1513232). All suggested corrections and changes are marked with track changes in the main manuscript. All suggested corrections and changes are marked with track changes in the main manuscript. In short, the gold standard of the study was defined more precisely, additional studies were introduced and discussed as proposed, a study flow diagram modified according to STARD criteria was introduced and the STROBE guidelines were applied to improve the quality of the manuscript. A detailed point-by-point rebuttal letter for each referee was provided. In addition, professional language editing was performed as recommend.

All authors involved in this work would like to thank the editors and referees for their time and effort since their suggested changes substantially improved the quality of the manuscript. Thank you very much for reconsidering our manuscript for publication in Cancers.

Referee 1

1.1.The authors present a study on radiomics using 30000 features to evalaute lymph nodes in head and neck carcinoma patient cohort. It is clinically very interesting and valuable topic. The results are nicely shown with numerous figures.

All authors involved very much appreciate the positive feedback of referee 1 about the topic and presentation of the present study.  

1.2. “Design of the study misses what is used as a golden standard. Is there a histological or cytological verification of the results? Please expand this issue and explain it in details.”

The authors very much apologize for the imprecision in the Methods section of the manuscript and strongly appreciate this as constructive suggestion. The definition of the gold standard used in the present study was indeed not defined clearly enough in the previous version of the manuscript.

All patients enrolled in the present study were treated with primary concurrent radiochemotherapy. The histopathologic confirmation of the diagnosis of HNSCC was performed by obtaining tissue samples from the primary tumor during staging panendoscopy. No additional ultrasound-guided core-needle biopsies were performed except from four patients diagnosed with carcinoma of unknown primary. In these four patients an ultrasound-guided core-needle biopsy of the largest suspect cervical lymph node was performed to confirm the diagnosis.

Although this diagnostic approach is in line with the recommendations by the National Comprehensive Cancer Network from a research perspective a histological or cytological verification of the segmented LNs would be improve the quality of the study. However, for ultrasound-guided core-needle biopsies the risk of bleeding as well as the risk of tumor seeding was considered too high, especially since a total of three biopsies per patient and CT-scan would have been required to define the histopathologic result per lymph node as reference. In addition, ultrasound guided core-needle biopsies usually do not aid in the distinction of “pathologic LNs” and “pathologic LNs with extracapsular spread”.

Alternatively, the option of performing ultrasound-guided fine-needle aspirations of the suspect cervical LNs was discussed. For this method, the risk of bleeding and tumor seeding was observed to be significantly lower than for core-needle biopsies. However, the number of ultrasound-guided fine-needle aspirations per patient would remain the same and non-diagnostic aspirations as well as inconclusive cytological findings have been previously reported.

Consequently, we refrained from this ultrasound-guided core needle biopsies or fine-needle aspirations. Instead, the classification of the segmented LNs in “pathologic with extracapsular spread”, “pathologic” and “non-pathologic” by the two experienced head-and-neck-radiologists based on established criteria [3-5] was defined as reference method. Although this approach bears the risk of incorrect classification, it was considered the safest approach for the patients. This approach was also recommended by the local ethics committee.

All authors involved in the preparation of this manuscript are aware of this major limitation. Consequently, numerous changes were made accordingly to the entire manuscript. Please refer to the Abstract, line 33-34, the Methods section, line 194-207 and the Discussion section, line 501-528.

1.3. “Discussion is quite short and should be expanded."

All authors apologize for the short discussion. Since referee 1 raised a very valuable concern about the imprecise definition of the reference method of the present study, this topic as well as its implication as major limitation of this study was added to the discussion section of the manuscript. In addition, the key results of the study, further limitations of the study, a cautious interpretation of the results and the generalizability of the results were added to the Discussion section, as recommended.

Please refer to the Discussion section, line 420-439, line 490-494 and line 501-552.

Reviewer 2 Report

  • line 51, The specific lymph node involvement at the laterocervical level correlates with the staging of the tumor, the patient's prognosis and the treatment administered, providing a better prognosis in related HPV carcinomas even with robotic surgery. please discuss and cite doi:10.1016/j.anl.2021.05.007
  • Machine learning models identify patients with intermediate-risk who could benefit from chemoradiation. These models predicted that approximately half of such patients have no added benefit from chemotherapy. please discuss and cite doi:10.1001/jamanetworkopen.2020.25881
  • please report a flow diagram with the study protocol to describe the selection criteria.
  • apply the strobe guidelines to the paper to improve the quality

Author Response

Dear editors, dear referees,

thank you very much for the extension of the revision deadline over the Christmas holidays. All the authors very much appreciate the patience of both, editors and referees. The authors very much appreciate the constructive and positive comments of both referees about the manuscript “Benchmarking eliminative radiomic feature selection for head and neck lymph node classification” (ID cancers-1513232). All suggested corrections and changes are marked with track changes in the main manuscript. All suggested corrections and changes are marked with track changes in the main manuscript. In short, the gold standard of the study was defined more precisely, additional studies were introduced and discussed as proposed, a study flow diagram modified according to STARD criteria was introduced and the STROBE guidelines were applied to improve the quality of the manuscript. A detailed point-by-point rebuttal letter for each referee was provided. In addition, professional language editing was performed as recommend.

All authors involved in this work would like to thank the editors and referees for their time and effort since their suggested changes substantially improved the quality of the manuscript. Thank you very much for reconsidering our manuscript for publication in Cancers.

Referee 2

2.1. “line 51, The specific lymph node involvement at the laterocervical level correlates with the staging of the tumor, the patient's prognosis and the treatment administered, providing a better prognosis in related HPV carcinomas even with robotic surgery. please discuss and cite doi:10.1016/j.anl.2021.05.007”

The authors very much appreciate this valuable suggestion. The paper mentioned by referee 2 was cited and discussed in the manuscript, as suggested. Please refer to the Introduction section, line 48-54, the Discussion section, line 420-424 as well as the Reference Section, line 581-584.

2.2. “Machine learning models identify patients with intermediate-risk who could benefit from chemoradiation. These models predicted that approximately half of such patients have no added benefit from chemotherapy. please discuss and cite doi:10.1001/jamanetworkopen.2020.25881”

The authors very much appreciate this valuable suggestion. The paper mentioned by referee 2 was cited and discussed in the manuscript, as suggested. Please refer to the Introduction section, line 78-82, the Discussion section, line 427-429 as well as the Reference Section, line 594-595.

2.3. “please report a flow diagram with the study protocol to describe the selection criteria.”

The authors very much appreciate this valuable suggestion. The study flow and patient inclusion was provided as flow diagram modified according to STARD criteria. Please refer to the Methods section, line 117-125, the Reference section of the manuscript, line 604-605 and the new figure 1.

2.4.apply the strobe guidelines to the paper to improve the quality”

The authors are very thankful for this valuable input. The STROBE Guidelines ware applied to the manuscript to improve the quality of the paper, as requested. Please refer to the Simple Summary section, line 17, the Abstract section, line 31 and line 33-34, the Introduction section, line 98-100, the Methods section, line 106-107, line 117-124, line 194-207 and line 222-225, the Discussion section, line 420-439, line 490-494, line 501-552 as well as the Reference section, line 602-611 and line 637-642. Please also refer to the new figure 1. 

Round 2

Reviewer 1 Report

The authors have improved the paper. It is well written, planned and discussed with illustrative presentation of results.